# The Enhancement of the Photodynamic Therapy and Ciprofloxacin Activity against Uropathogenic *Escherichia coli* Strains by *Polypodium vulgare* Rhizome Aqueous Extract

**DOI:** 10.3390/pathogens10121544

**Published:** 2021-11-25

**Authors:** Dorota Tichaczek-Goska, Michał Gleńsk, Dorota Wojnicz

**Affiliations:** 1Department of Biology and Medical Parasitology, Wrocław Medical University, 50-367 Wrocław, Poland; dorota.tichaczek-goska@umw.edu.pl; 2Department of Pharmacognosy and Herbal Medicines, Wrocław Medical University, 50-367 Wrocław, Poland; michal.glensk@umw.edu.pl

**Keywords:** photodynamic therapy (PDT), chlorin e6, *Polypodium vulgare* rhizome, plant extract, uropathogenic *Escherichia coli*, ciprofloxacin, biofilm

## Abstract

Antibiotic therapy and photodynamic therapy (PDT) are commonly used to treat bacterial infections. Unfortunately, these methods are often ineffective. Therefore, agents that could effectively support antibiotic therapy and PDT in the inactivation of pathogens are being sought. Phytotherapy seems to be a good solution. The aim of the current research was to examine whether *Polypodium vulgare* extract (PvE) would improve the effectiveness of PDT and ciprofloxacin (CIP), an antibiotic that is commonly used to treat urinary tract infections in humans. UHPLC-MS analysis was performed to establish the PvE content. Chlorin e6 has been used as a photosensitizer in the PDT method. Biofilm production was established using the spectrophotometric method. The live cell count in planktonic and biofilm consortia was determined with the microdilution method and DAPI staining. The decrease of the bacterial survival, biofilm mass synthesis, and morphological changes of the bacteria under the combined treatments: PDT+PvE and CIP+PvE was noted. The results clearly indicate that the PvE can be used as a good agent for improving the efficacy of both PDT and the CIP activity to inactivate uropathogenic *Escherichia coli* strains. The obtained results are of particular value in the era of widespread and still-increasing drug resistance among bacterial pathogens.

## 1. Introduction

The growing resistance of pathogenic bacteria to commonly used antibiotics [1,2] and frequent failures in the use of these chemotherapeutics in the treatment of bacterial infections has led to the search for alternative methods and agents that exhibit high antimicrobial activity. One such alternative antimicrobial method is photodynamic therapy (PDT) [3]. PDT was first described at the beginning of the twentieth century and since then has been successfully used in various branches of medical sciences [4,5]. PDT involves an interaction between the molecules of a photosensitizer and light of the proper dose and wavelength which proceeds to produce a reactive oxygen species that is able to quickly destroy cell membranes, damage DNA, and/or oxidize a cell’s organic compounds [6,7]. PDT is non-invasive, and, in addition, the light stream can be focused on the target area thus avoiding the irradiation of healthy locations. Thus, PDT has been used in various medical procedures, such as dental procedures [8], dermatological problems [9,10], or in cancer treatment [11,12]. PDT is also used to inactivate different bacterial pathogens including uropathogenic strains [13,14,15]. *Escherichia coli* is known as the most prevalent Gram-negative bacteria that is responsible for urinary tract infections (UTIs) in humans [16]. UTIs belong to the group of infectious diseases that most frequently affect the human population. Although the etiology of UTIs itself has not changed significantly for years, strains with increasing drug-resistance are being isolated more frequently which is a very serious clinical problem faced by today’s medicine. According to the European Association of Urology guidelines on urological infections, ciprofloxacin (CIP) is recommended for empirical antimicrobial therapy in UTIs, including pyelonephritis [17]. Its therapeutic function is related to the inhibition of DNA gyrase, which prevents bacterial DNA replication and thus cell division. The increasing number of recurrent UTIs [18,19] and the overuse of antibiotics has led to the selection of multi-drug resistant bacterial strains. PDT is recommended for the treatment of infections that are caused by multidrug-resistant bacteria. However, this therapy is often ineffective, especially in the case of Gram-negative bacteria due to the complex structure of their envelope. Therefore, new methods and remedies that could support PDT are constantly being searched for. More and more attention is being paid to natural remedies that were used in the past and nowadays in folk medicine [20]. One such natural remedy that ids used in UTIs is *Polypodium vulgare* L. (Polypodiaceae). In traditional Polish folk medicine, polypody fern rhizome infusions were used as a diuretic agent and were especially used in the treatment of chronic nephritis and pyelonephritis [21,22]. Although the *P. vulgare* rhizome extract (PvE) exhibits multiple, well studied pharmacological effects, e.g., diuretic, antioxidant, cancerostatic, and is in common use, knowledge about its antibacterial properties is still quite limited [23,24,25,26,27,28].

To the best of our knowledge, there are no studies on the antibacterial interaction of the common fern rhizome extract and PDT. For the above reasons, the aim of this study was to investigate the antibacterial properties of PvE against uropathogenic *E. coli* strains and to establish whether it can enhance the chlorin e6-mediated photodynamic reaction. We also decided to investigate whether PvE can work synergistically with ciprofloxacin against uropathogenic *E. coli* strains

## 2. Results

### 2.1. P. vulgare Extract Quantitative Analysis

Based on the MS spectra, the most prominent peak (6) belonged to osladin with *m*/*z* [M − H]¯ at 885 and to a lesser extent to polypodoside A with *m*/*z* [M − H]¯ at 883. Peaks 1, 2, and 3 were characterized as catechin derivatives with *m*/*z* [M − H]¯ 421, whereas peak 4 was afzelechin and its derivatives with *m*/*z* [M − H]¯ at 405. Finally, peak 5 was tentatively characterized as a mixture of hydroxyecdysone at *m*/*z* [M − H]¯ 479 and polypodine B at *m*/*z* [M − H]¯ 495 [29,30] (Figure 1).

### 2.2. Antibacterial Effect of PDT and P. vulgare Extract

In the first step of the study, the effect of PDT with the use of chlorin e6 on the survival of uropathogenic *E. coli* strains was determined. As can be seen in Figure 2 and Figure 3, the survival of both of the *E. coli* strains (reference CFT073 and clinical) decreased slightly right after irradiation (t_0_) and 1 h (t_1_) after PDT treatment (Ce6-L). However, these results were not statistically significant (*p* > 0.05). It should be added that the chlorin e6 itself (without light exposure) and the light exposure itself (without the photosensitizer) did not cause side effects in the form of a decrease in the cell count rate of the investigated strains (data not shown).

Therefore, it was investigated whether the overnight incubation of bacterial strains in the presence of PvE would improve the antibacterial effect of the chlorin-e6-induced photodynamic reaction. The results that are presented in Figure 2 indicate that the reference *E. coli* CFT073 strain had become susceptible to PDT (PvE+Ce6-L). A statistically significant decrease in the live cell count of bacteria (PvE+Ce6-L) in relation to the irradiated (Ce6-L), was recorded immediately after irradiation (t_0_), and was prolonged to 4 h after light exposure (t_0_–t_4_) (*p* ≤ 0.05). The greatest decreases in the number of viable cells were recorded at t_0_, t_1_, and t_2_ and it was 73%, 70%, and 76%, respectively. 

The analysis of the data that is contained in Figure 3 shows that the combined action of pre-incubation of bacteria with PvE followed by chlorin e6-induced PDT (PvE+Ce6-L) on the clinical *E. coli* strain was slightly less effective. The significant antibacterial effect lasted up to 3 h after irradiation (t_0_–t_3_). The reduction of the viable cell count ranged from 77% at time t_0_ to 52% at time t_3_ (*p* ≤ 0.05).

### 2.3. Antibacterial Effect of CIP and P. vulgare Extract on Planktonic Cultures

In the next step of the research, the antibacterial effect of CIP on the survival of planktonic forms of both the studied UPECs was determined. As shown in Figure 4, CIP did not significantly reduce the number of bacterial cells (*p* > 0.05) compared to the control samples except for the 2-h and 4-h culture of the reference CFT073 strain (Figure 4A) (*p* ≤ 0.05). On the other hand, the presence of the PvE significantly reduced the number of live bacteria of both of the tested *E. coli* strains (Figure 4A,B). Therefore, it was examined whether the incubation of *E. coli* rods in a medium that contained both CIP and PvE would result in reduced survival of these strains in comparison to their survival when incubated with CIP alone. The results that are presented in Figure 4 show the synergistic effect of the drug and the plant extract on both of the bacterial strains at all stages of their cultivation. The percentage decrease of viable cell counts of strain CFT073 ranged from 73–49% in the samples containing CIP+PvE mixture (Figure 4A). Similar results were obtained for the clinical *E. coli* strain (73%–53% reduction) (Figure 4B).

### 2.4. Antibacterial Effect of CIP and P. vulgare Extract on Biofilm Cultures

In the human body, bacteria often form biofilm consortia which are more resistant to the action of antibiotics as compared to planktonic forms. Therefore, the next step of the research was to investigate whether there is a synergism of action of CIP+PvE combination against UPECs that are living in biofilm consortia. Based on the results in Figure 5A,B, it can be concluded that the clinical *E. coli* strain was a better biofilm producer than the reference CFT073 strain. The clinical strain produced a moderate biofilm (OD > 0.084) at all stages of development (24–72 h), whereas the reference strain developed a moderate biofilm only after 72 h of cultivation (Figure 5A). As shown in Figure 5, CIP slightly decreased the amount of biofilm that was created by the strains, however, these results were not statistically significant (*p* > 0.05). The much better antibacterial activity was noted when the bacteria were incubated with PvE (*p* ≤ 0.05). In the case of the reference CFT073 strain, the extract completely inhibited biofilm synthesis in the older 48-h and 72-h consortia. The clinical *E. coli* strain from a moderate biofilm producer turned into a weak biofilm producer under the influence of PvE. The combination of CIP+PvE completely inhibited biofilm synthesis by the reference CFT073 strain, regardless of the incubation time (*p* ≤ 0.05), indicating the synergism of the action of these two antimicrobials (Figure 5A). It is worth emphasizing that the complete inhibition of biofilm mass production by the clinical *E. coli* strain occurred only under the influence of the combination of CIP+PvE in mature 48-h and 72-h cultures (Figure 5B).

In the study, not only the influence of CIP and PvE on the amount of produced biofilm mass but also the influence of these two antimicrobials on the survival of the bacteria in the biomass was examined. Both CIP and PvE, when used separately, reduced the survival of the reference CFT073 strain, but it was statistically insignificant (Figure 6A). Interestingly, the survival of the clinical *E. coli* strain decreased significantly under the influence of CIP as well as PvE but only in the 24-h and 48-h biofilms (*p* ≤ 0.05) (Figure 6B). The most effective anti-growth activity was noted in biofilms that were subjected to the simultaneous treatment of the CIP+PvE combination compared to the untreated bacteria for both of the tested strains (*p* ≤ 0.05). The results also indicate that the anti-growth effect of CIP+PvE was more effective as compared to the action of CIP that was used alone. Such synergism was evident in the 72-h culture of the reference CFT073 strain (62% reduction; Figure 6A) and in the 48- and 72-h cultures of the clinical *E. coli* strain (73% and 81% reduction, respectively; Figure 6B) (*p* ≤ 0.05).

Morphological changes of the bacterial cells of both of the tested strains under the influence of PvE and CIP were visualized by fluorescence microscopy (Figure 7). In the presence of PvE, the bacterial cells were slightly longer (short filaments; 5–15 µm) than in the controls, while in the cultures that contained CIP+PvE, the additional long filaments (>15 µm) was observed (Figure 7).

## 3. Discussion

Since ancient times, *P. vulgare* rhizome has been used in traditional European, American, and Chinese medicine, mainly as a remedy for respiratory diseases [20,27,31]. In traditional Polish medicine, polypody fern rhizome infusions were used as an expectorant and diuretic and were used exclusively for treating chronic nephritis and pyelonephritis [21,22]. *P. vulgare* is very rich in biologically active compounds, which belong to different classes of plant metabolites: saponins, triterpenoids, ecdysteroids, tannins, flavonoids, and phenolics [20,29,30,32,33]. UHPLC-MS analysis of PvE that was used in our study revealed the presence mainly of osladin and polypodoside A belonging to saponins, flavonoids: catechin derivatives and afzelechin, and ecdysteroid hormones: hydroxyecdysone and polypodine B. As demonstrated by Rojewska et al. [34] saponins can interact with bacterial membranes and influence their physical properties, leading to changes in hydrophobicity of the cell surface. This can increase the uptake of antibiotics to bacterial cells and, consequently, improve their antibacterial activity. It has been shown that flavonoids, especially catechins, can damage bacterial membranes by interacting with the lipid bilayer [35,36]. This way flavonoids allow the penetration of the bacterial cell by antibiotics. Phytoecdysteroid hormones that are present in many plant extracts, including the PvE that was used in the current study, show the antibacterial properties that are associated with their antioxidant effect [37]. Due to their multifarious chemical structure and the presence of inter alia hydroxyl residues, plant secondary metabolites can easily incorporate into bacterial cell envelopes, changing their hydrophobicity, fluidity, and permeability. Such changes can improve the activity of various antibacterial agents (e.g., PDT, antibiotics). When the bacterial membrane is damaged, reactive oxygen species that are generated during PDT can penetrate more easily inside the cell and disrupt its metabolic processes. The changes in the cell membrane can additionally cause the outflux of cytoplasmic substances that are essential for the growth of the microorganism [38,39]. Studies on plant extracts and polyphenols that are isolated from them indicate that these compounds may interact synergistically with other antimicrobial agents, increasing their biocidal effect. 

Due to the above-described properties of the main components of the PvE, it was decided to investigate whether this extract could improve the activities of chlorin-e6-induced PDT and the commonly used antibiotic, ciprofloxacin, against UPEC strains. Many reports indicate the effectiveness of chlorin e6 in PDT against bacterial pathogens [40,41,42]. However, chlorin e6 is not always an effective photosensitizer, especially when applied against Gram-negative bacteria [13,43,44,45]. The ineffectiveness of PDT to Gram-negative bacteria is associated with the low permeability of their outer membrane for antibacterial agents. To improve the antimicrobial activity of photosensitizers, research groups are trying to synthesize their various derivatives, create conjugates, or use them in parallel with other compounds, e.g., antibiotics [13,43,44,46,47,48,49,50,51].

In our present work, as well as in the previous study [13], chlorin-e6-induced PDT showed very low antibacterial activity against the investigated *E. coli* strains. Park et al. [52] also investigated the antimicrobial effect of PDT using chlorin e6 against various pathogenic bacteria. To examine the antimicrobial effect of Ce6-mediated PDT against *S. aureus, Pseudomonas aeruginosa*, and *E. coli*, growth inhibition zones, CFU quantification and bacterial viability were evaluated. Indeed, the results that were obtained showed that Ce6-mediated PDT very effectively inhibited the growth and survival of *S. aureus* and *P. aeruginosa* but had only a weak effect on *E. coli*. However, the results that were obtained in the current study indicated that both the tested UPECs that were treated overnight with PvE became more susceptible to chlorin-e6-induced PDT as compared to the bacteria that were untreated with the extract. Accordingly, our work is innovative because it is the first that has investigated the use of plant extract as the agent that was enhancing the action of PDT with the use of the well-known photosensitizer (chlorin e6) against bacteria pathogenic to humans. It is difficult to discuss the results of our study in light of the results of other research groups. All of the available reports on alternative antibacterial agents, such as plant extracts, secondary metabolites, and PDT, are based on their use as photosensitizers [41,53,54]. Dascalu et al. [43] used gels containing three essential oils (EO) extracted from *Boswelia carteri*, *Origanum vulgare*, and *Curcuma longa* as photosensitizers against *E. coli* and *S. aureus* strains. The irradiation was performed using two sources of light—an LED and a laser diode. The antibacterial activity was examined by measuring of the diameter of the growth inhibition zones. The antibacterial activities of samples that contained *O. vulgare* and *B. carteri* EOs were similar and quite good for both of the strains with inhibition zones ranging from 8 mm to 12 mm. The largest difference between the diameters of the bacterial inhibition zone was recorded in the case of the sample containing *C. longa* EO. The extract was completely ineffective on Gram-negative *E. coli* rods but exhibited quite good activity against *S. aureus* cocci. The interesting finding was achieved by Yow et al. [53], who indicated that the simultaneous application of hypericin (naturally occurring polycyclic quinine obtained from *Hypericum perforatum* L.) and light irradiation causes a reduction in viability and significant changes in methicillin-sensitive and resistant *S. aureus* bacteria. However, the hypericin-induced PDT had no antibacterial effect on the growth of the *E. coli* strain. dos Santos et al. [55] conveys the use of *Myrciaria cauliflora* (Mart.) O. Berg extract as a photosensitizer which, when exposed to light, was active against *S. aureus*. The use of the extract and PDT efficiently decreased the survival of the investigated cocci. 

The second part of our research examined whether the PvE can improve the antibacterial activity of ciprofloxacin—an antibiotic commonly that is used in the treatment of UTIs’. When used alone, CIP did not reduce the number of bacteria as compared to the control samples, except for a young planktonic culture of the reference *E. coli* CFT073 strain. However, the results that were obtained showed the synergistic effect of the drug and the PvE on both the reference and clinical uropathogens at all of the stages of their planktonic cultivation. Similarly, only the combined action of PvE and CIP efficiently inhibited biofilm mass synthesis and decreased bacterial survival in biofilm consortia. The synergistic effect may be related to the blocking of efflux pumps by the plant compounds, which prevents the release of the antibiotic outside the cell and leads to inactivation or death of the bacteria [56]. It is known that plant-derived compounds may disrupt different mechanisms of bacterial resistance to conventionally used antibiotics. They can inhibit the modification of the cellular target of the antibiotic, facilitate the penetration of the antibiotic inside the cell or inhibit its leakage, and also can block bacterial antibiotic-utilizing enzymes [57]. The synergistic effect of plant extracts, essential oils, and plant-derived compounds with antibiotics e.g., ciprofloxacin, against pathogenic bacteria has been confirmed by many other researchers [58,59,60,61,62].

## 4. Materials and Methods

### 4.1. Microorganisms

A total of two uropathogenic *Escherichia coli* strains (UPEC) were used—reference *E. coli* CFT073 (ATCC 700928) strain from the American Type Culture Collection and clinical uropathogenic *E. coli* from the bacterial collection of the Department of Biology and Medical Parasitology, the Wroclaw Medical University, Poland. The strains were maintained on slopes containing a nutrient broth and glycerol in a final concentration of 40% and were stored at −20 °C. 

### 4.2. Antimicrobial Agents

#### 4.2.1. Antibiotic

Ciprofloxacin lactate (Proxacin^®^; Polpharma, Warsaw, Poland) was used in the experiment.

#### 4.2.2. Plant Material

*P. vulgare* rhizome was purchased from Alfred Galke GmbH (Flecken Gittelde, Germany) company. A voucher specimen numbered PV-1/2016 was deposited in the Herbarium of the Department of Pharmacognosy and Herbal Medicines of the Wroclaw Medical University.

#### 4.2.3. Photosensitizer and Light Source

Chlorin e6 (Ce6) was purchased from Frontier Scientific (Porphyrin Products, Logan, UT, USA) and dissolved in DMSO and sterile deionized water (1:1) to obtain a stock concentration of 500 µg/mL. Finally, the Ce6 concentration of 100 µg/mL was then used to measure the photodynamic efficiency against the *E. coli* strain. Diode laser LASER COUPLER 635 (UniTech, Wroclaw, Poland) was used as a light source (wavelength 635 nm) to irradiate Ce6 at the power density of 290 mW/cm^2^ and total energy dose of 120 J/cm^2^ with no thermal side effects [13].

### 4.3. P. vulgare Rhizome Aqueous Extract Preparation

A total of 620 g air-dried rhizomes were extracted with water by macerating the material in three rounds (24 h using 2500 mL of water for each). Following filtration and solvent evaporation, the aqueous extract yielded 124 g (~20%) of residue.

### 4.4. P. vulgare Rhizome Aqueous Extract Quantification

UHPLC-MS was performed according to Gleńsk et al. [23].

### 4.5. MIC Determination

Minimum inhibitory concentrations (MICs) of CIP and PvE were performed in Mueller-Hinton broth (MHB; Gdansk, Poland) according to CLSI guidelines for broth microdilution susceptibility testing [63]. In the current study, the MICs of CIP were 0.0039 µg/mL and 0.007 µg/mL for clinical *E. coli* and reference *E. coli* CFT073, respectively. The MIC of PvE was 60 mg/mL for both of the analyzed UPEC strains.

### 4.6. Preparation of Bacterial Suspensions

Bacterial strains grown overnight on MHB (Emapol, Gdansk, Poland) at 37 °C, were transferred to fresh MHB, and incubated at 37 °C for 2 h in the shaking water bath Julabo SW-22 (DanLab, Bialystok, Poland). The bacterial suspensions were then centrifuged and suspended in PBS to reach the final density of 0.5 McFarland. The suspensions that were prepared in this way were used in all of the experiments.

### 4.7. Planktonic Bacterial Cultures Assay

Bacterial suspensions were cultured in Eppendorf tubes at 37 °C for 2-, 4-, 6-, and 24-h in the MHB medium containing CIP at a concentration of 0.25 × MIC or PvE at 0.75 × MIC, or both agents at concentrations that were mentioned above. The untreated (control) samples did not contain any antibacterial agent.

### 4.8. Biofilm Assay

Cultures were carried out in 96-well polystyrene microtiter plates for 24-, 48-, and 72 h. The control biofilm samples did not contain any antibacterial agent, while the test samples contained CIP at a concentration of 0.5 × MIC or PvE at 0.75 × MIC, or both agents at mentioned above concentrations. Unbound cells were removed from the 24-, 48-, and 72-h biofilm cultures by gentle washing three times with sterile PBS. Then, 1% crystal violet was added to each well and incubated at 37 °C for 15 min to penetrate the CV into the biofilm structure. Next, the dye was removed and the crystal violet was eluted from the biofilm mass by 95-% ethanol. The optical density (OD) was measured at a wavelength of 590 nm in a microplate reader (HiPo MPP-96^®^ BIOSAN, Kraków, Poland). On the basis of the OD values that were obtained, the bacteria were classified into one of the following groups: OD ≤ ODc—non-biofilm producer; ODc < OD ≤ 2 × ODc—weak-biofilm producer; 2 × ODc < OD ≤ 4 × ODc—moderate-biofilm producer; and 4 × ODc < OD—strong-biofilm producer. The ODc value was calculated as the sum of the mean OD values for the blank (MHB) and 3 times the standard deviation of the mean OD value for the MHB [64,65]. In current study the ODc was 0.042.

### 4.9. Determination of Bacterial Survival

Viable counts of UPECs growing in different conditions (planktonic cultures and biofilms that were treated with antibacterial agents and untreated) were determined by the serial dilution method. The agar plates were incubated at 37 °C for 24 h and the bacterial colonies were counted and reported as colony forming units per milliliter (CFU/mL).

### 4.10. PDT Experimental Conditions

Cultured overnight bacterial suspensions (1–2 × 10^7^ CFU/mL) that were treated with PvE at a concentration of 0.5 × MIC or the untreated (control) were plated to the wells of a 96-well plate. Then the Ce6 stock solution was also added to the wells to obtain the final concentration of 100 µg/mL of the photosensitizer. The plate was preincubated for 15 min at room temperature (in the dark) before exposure to the red light at 120 J/cm^2^. The samples were then diluted and cultured in triplicate on nutrient agar plates (Biomed, Kraków, Poland) immediately (t_0_), at 1-(t_1_), 2-(t_2_), 3-(t_3_), and 4-h (t_4_) after irradiation. The agar plates were incubated at 37 °C for 24 h and the number of CFU/mL was counted. The following experimental groups were used in the study: (i) bacteria+PvE, (ii) bacteria+Ce6 and red laser light (Ce6-L), (iii) bacteria+PvE+Ce6 irradiated with red laser light (PvE+Ce6-L), and the control sample containing no plant extract nor being irradiated (untreated).

### 4.11. Biofilm Visualization by DAPI Staining

For visualization, the biofilms were allowed to grow on polystyrene pieces (0.5 × 0.5 cm) that were placed in PvE at 0.75 × MIC, CIP at 0.5 × MIC and the combination CIP+PvE. After 24-, 48- and 72 h at 37 °C the polystyrene pieces were washed with PBS and stained with 1% DAPI solution. After 10 min of staining in the dark, the DAPI solution was removed by rinsing with PBS. The polystyrene pieces with biofilm were dried in the dark at room temperature and were analyzed by fluorescence microscopy with an Eclipse 400 (Nikon, Tokyo, Japan). The biofilms were analyzed under 1000-fold magnification.

### 4.12. Statistical Analysis

Each experiment was repeated three times (so-called technical repeat); the grown bacterial colonies were counted from 6–8 plates, and the optical density was read from 6 wells of the microtiter plate, which gave a total of 18 biological repeats. The final results are average values. The differences in the number of the viable bacteria and biofilm formation that were exposed to various antibacterial agents, their combinations, and unexposed samples were analyzed using Kruskal-Wallis’ non-parametric test followed by a Dunn’s multiple comparison test. Statistical calculations were made using Statistica 13.3. (Stat Soft, Kraków, Poland). Values of *p* ≤ 0.05 were considered statistically significant.

## 5. Conclusions

The results of the present study are promising because they clearly indicate that the *P. vulgare* rhizome aqueous extract can be used as a good agent for improving the efficacy of both photodynamic therapy and the activity of ciprofloxacin to inactivate UPEC strains. The obtained results are of particular value in the era of widespread and still-increasing drug resistance among pathogenic bacteria. There is no doubt that further, more detailed research is required to broaden our knowledge in this field.

## Figures and Tables

**Figure 1 pathogens-10-01544-f001:**
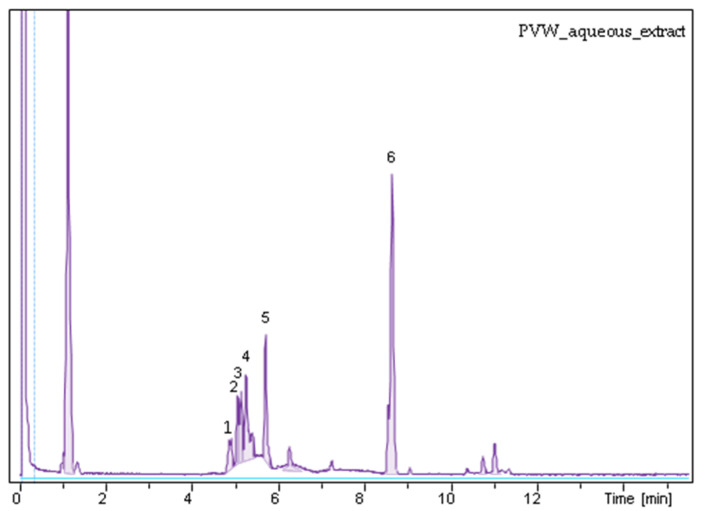
UHPLC-MS chromatogram of *P. vulgare* rhizome aqueous extract.

**Figure 2 pathogens-10-01544-f002:**
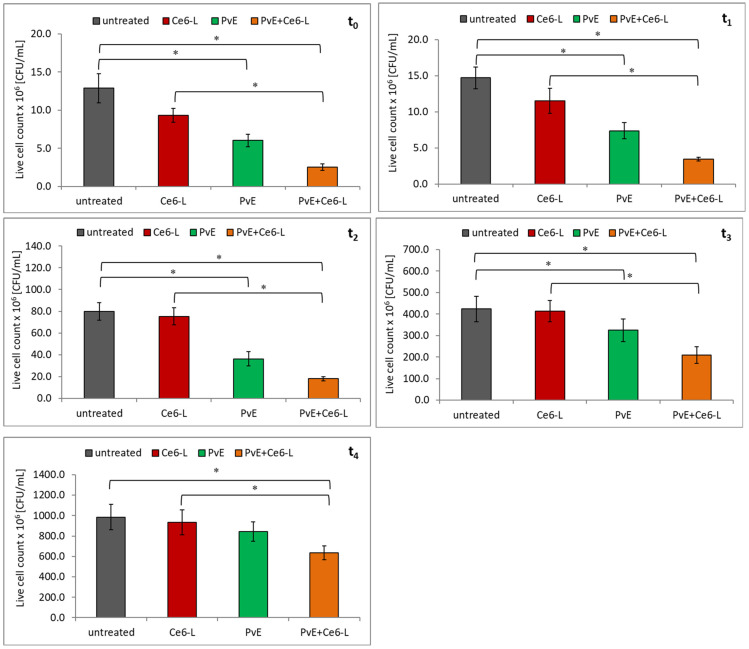
The influence of the chlorin-e6-induced PDT (Ce6-L), *P. vulgare* rhizome extract (PvE), and the combination PvE+Ce6-L on the survival of the reference *E. coli* CFT073 strain; untreated (control)—no PvE, no irradiation; statistically significant differences were noted with an asterisk (* *p* ≤ 0.05).

**Figure 3 pathogens-10-01544-f003:**
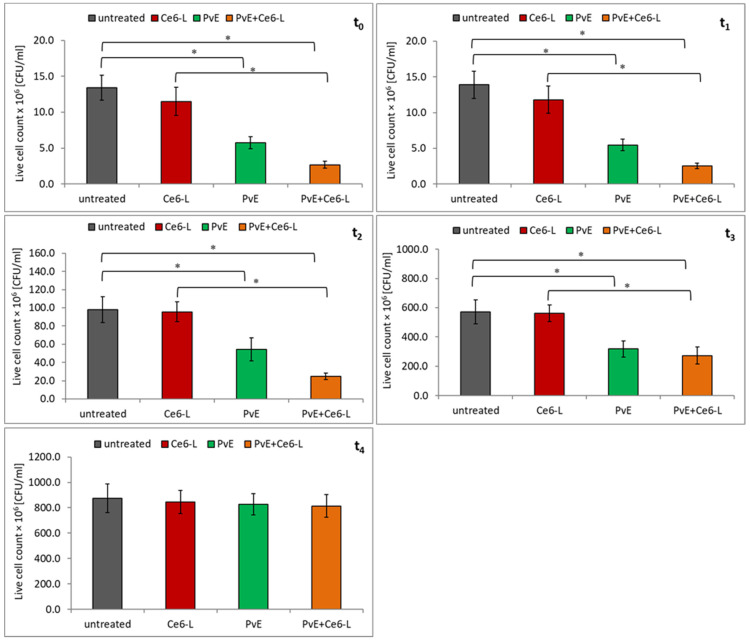
The influence of the chlorin-e6-induced PDT (Ce6-L), *P. vulgare* rhizome extract (PvE), and the combination PvE+Ce6–L on the survival of the clinical uropathogenic *E. coli* strain; untreated (control)—no PvE, no irradiation; statistically significant differences were noted with an asterisk (* *p* ≤ 0.05).

**Figure 4 pathogens-10-01544-f004:**
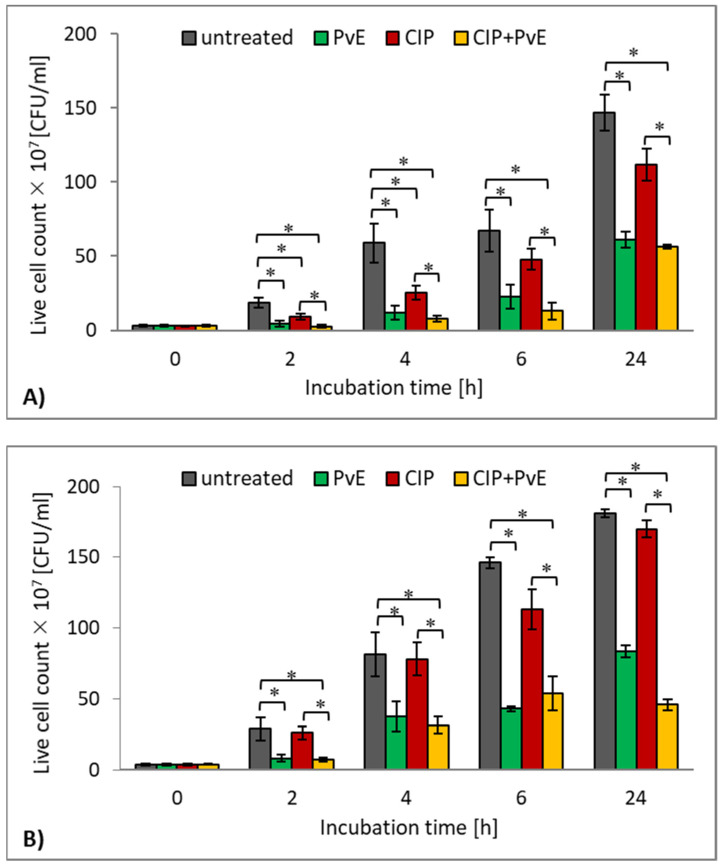
The time-dependent survival rate of the planktonic cultures of the reference *E. coli* CFT073 strain (**A**) and clinical *E. coli* strain (**B**) treated with ciprofloxacin (CIP), *P. vulgare* rhizome extract (PvE) and the combination CIP+PvE; untreated (control)—no PvE, no CIP; statistically significant differences were noted with an asterisk (* *p* ≤ 0.05).

**Figure 5 pathogens-10-01544-f005:**
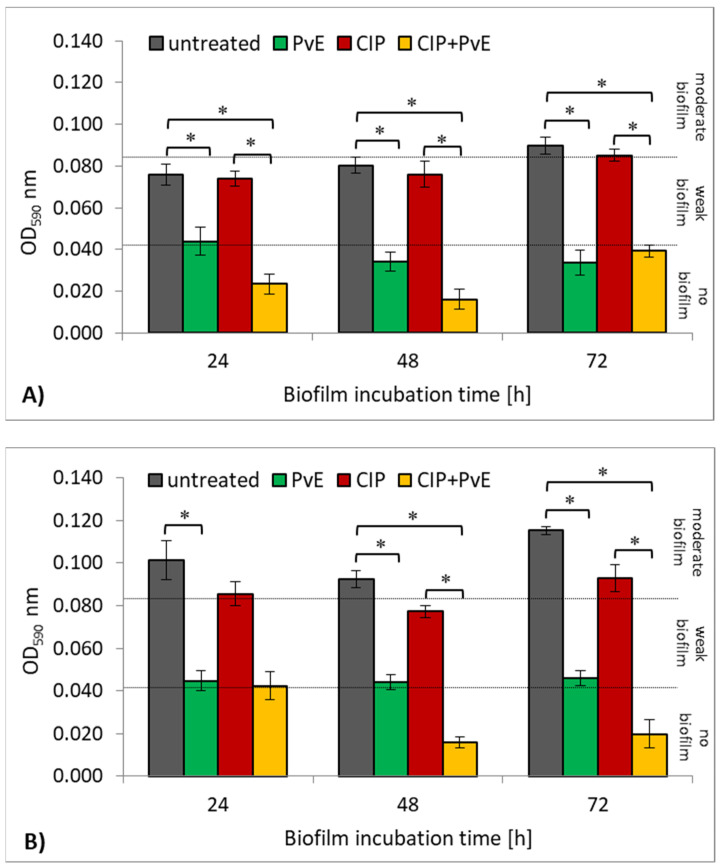
The amount of biofilm mass that was produced by the reference *E. coli* CFT073 strain (**A**) and clinical *E. coli* strain (**B**) treated with ciprofloxacin (CIP), *P. vulgare* rhizome extract (PvE) and the combination CIP+PvE; untreated (control)—no PvE, no CIP; statistically significant differences were noted with an asterisk (* *p* ≤ 0.05).

**Figure 6 pathogens-10-01544-f006:**
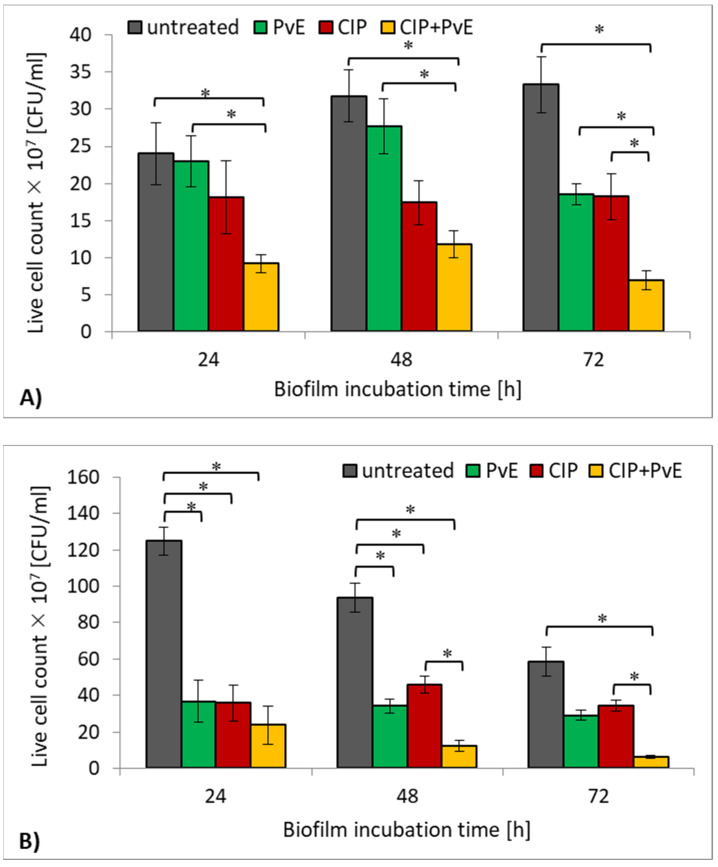
The survival rate of the reference *E. coli* CFT073 strain (**A**) and clinical *E. coli* strain (**B**) treated with ciprofloxacin (CIP), *P. vulgare* rhizome extract (PvE) and the combination CIP+PvE; untreated (control)—no PvE, no CIP; statistically significant differences were noted with an asterisk (* *p* ≤ 0.05).

**Figure 7 pathogens-10-01544-f007:**
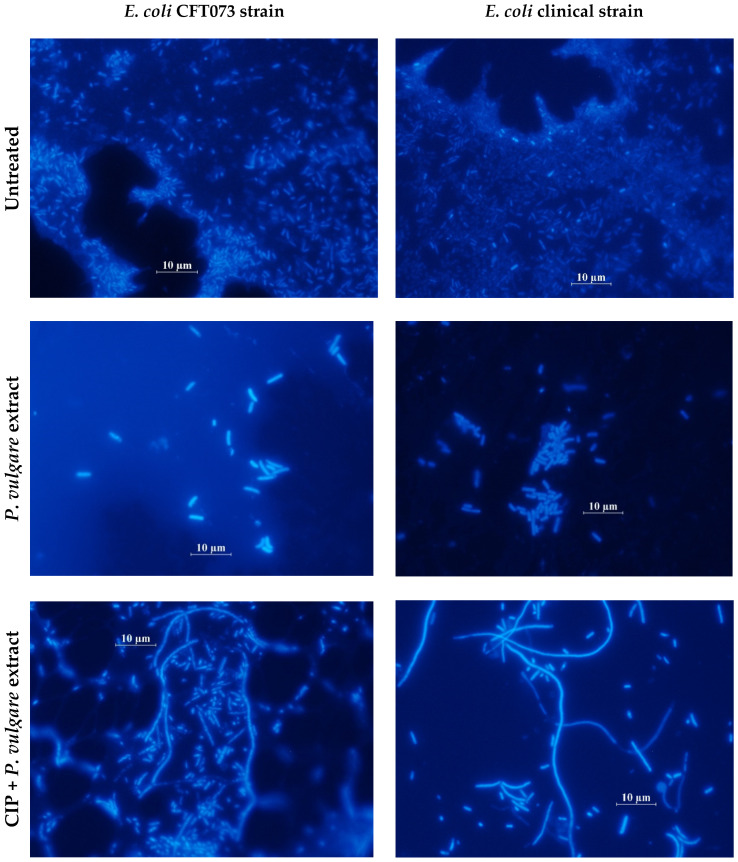
The examples of fluorescence microscopy images revealing the changes in cell morphology and biofilm-mass formation by the reference *E. coli* CFT0373 (**A**) and clinical *E. coli* (**B**) strains that were treated with *P. vulgare* extract and ciprofloxacin (CIP) (DAPI staining; magnification ×1000).

## Data Availability

The data presented in this study are available on request from the corresponding author.

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
