# Peer review of "The Enhancement of the Photodynamic Therapy and Ciprofloxacin Activity against Uropathogenic Escherichia coli Strains by Polypodium vulgare Rhizome Aqueous Extract"

_pathogens, 2021, doi:10.3390/pathogens10121544_

Round 1

Reviewer 1 Report

The authors demonstrated synergistic effect of ciprofloxacin and polypodium vulgare rhizome extract in UTI in the in vitro setting. The study is well designed as it compares the experimental arm to the non-treated arm. This study is of interest due to the potential benefits of this combination therapies in the practical setting. The authors however will need to address whether the polypodium vulgare rhizome extract is also beneficial in combination with other commonly used antibiotics besides ciprofloxacin. Also, how could this experimental study be made practical such as is this combination can be provided as a pill form so as to be administered orally in treatment of UTI. 

Author Response

REVIEWER COMMENTS:

The authors demonstrated synergistic effect of ciprofloxacin and polypodium vulgare rhizome extract in UTI in the in vitro setting. The study is well designed as it compares the experimental arm to the non-treated arm. This study is of interest due to the potential benefits of this combination therapies in the practical setting. The authors however will need to address whether the polypodium vulgare rhizome extract is also beneficial in combination with other commonly used antibiotics besides ciprofloxacin. Also, how could this experimental study be made practical such as is this combination can be provided as a pill form so as to be administered orally in treatment of UTI.

ANSWER:

Thank you very much for your time and all your valuable suggestions.

We would like to start our explanations with the fact that in the available literature there are no studies on the interaction of Polypodium vulgare extract and antibiotics.

We are a research group that studies the effect of various antibacterial agents (antibiotics, plant extracts, plant metabolites, photodynamic therapy) on bacteria isolated from the urine of patients with urinary tract infections. We have attached a few of our publications on this subject below.

  • Wojnicz et al. Is it worth combining Solidago virgaurea extract and antibiotics against uropathogenic Escherichia coli rods? An in vitro model study. Pharmaceutics 2021
  • GleÅ„sk et al. Differing antibacterial and antibiofilm properties of Polypodium vulgare rhizome aqueous extract and one of its purified active ingredients-osladin. J Herb Med 2019
  • Tichaczek-Goska et al. Photodynamic enhancement of the activity of antibiotics used in urinary tract infections. Lasers Med Sci. 2019
  • Wojnicz D et al. Pentacyclic triterpenes combined with ciprofloxacin help to eradicate the biofilm formed in vitro by Escherichia coli. Indian J Med Res 2015
  • Wojnicz et al. Effect of sub-minimum inhibitory concentrations of ciprofloxacin, amikacin, and colistin on biofilm formation and virulence factors of Escherichia coli planktonic and biofilm forms isolated from human urine. Braz J Microbiol 2013

In the above-listed studies, we used antibiotics recommended for the treatment of UTIs - usually ciprofloxacin and amikacin, and sometimes also colistin. The manuscript submitted for review contains only preliminary research results. Based on the results obtained in our previous studies, in which we demonstrated a better antibacterial effect of ciprofloxacin than amikacin and colistin - including their combination with PDT or with Solidago virgaurea extract, we first decided to check the interaction of Polypodium vulgare extract with ciprofloxacin. Our research plans will include experiments with other antibiotics and a wider range of clinical uropathogenic bacterial strains.

Our experimental study can be made practical. Currently, Polypodium vulgare is used in the form of rhizome infusion, inter alia, in inflammation of the urinary bladder. Theoretically, one could try to prepare and administer P. vulgare rhizome extract in the form of oral capsules. This form of the extract is used in the case of another fern - Polypodium leucotomos - used as an antioxidant, immune-boosting, and anti-aging agent.

Reviewer 2 Report

1. It would be advisable to study the urine metabolome of people taking the polypodium vulgare rhizome aqueous extract. Additionally, the physicochemical parameters of the urine of people consuming the polypodium vulgare rhizome aqueous extract should be determined. This would allow to say something about the mechanism of action.

2. Biofilm analysis using CV staining is not a valid method. It often leaches out less structured biofilm. It would be better to continuously monitor the biofilm formation process by microscopy, even without the need for fluorescent staining.

3.The coloring of the diagrams is a bit extravagant for publications.

Author Response

Thank you very much for your time and all your valuable suggestions.

REVIEWER COMMENT:  It would be advisable to study the urine metabolome of people taking the polypodium vulgare rhizome aqueous extract. Additionally, the physicochemical parameters of the urine of people consuming the polypodium vulgare rhizome aqueous extract should be determined. This would allow to say something about the mechanism of action.

ANSWER: We agree with this suggestion. The manuscript submitted for review contains only preclinical in vitro studies. In the next step, we will try to make clinical trials involving patients with recurrent urinary tract infections who will supplement the infusion of P. vulgare rhizomes. As we are employees of non-clinical research departments, this step will require cooperation with clinicians. Indeed, determination of the metabolome and physicochemical parameters of urine samples will contribute to broadening the knowledge about the impact of plant extract on the physical and biochemical parameters of urine. We consider it a great idea that we will try to implement in the nearest future.

REVIEWER COMMENT:  Biofilm analysis using CV staining is not a valid method. It often leaches out less structured biofilm. It would be better to continuously monitor the biofilm formation process by microscopy, even without the need for fluorescent staining.

ANSWER: We agree that the use of the crystal violet biofilm method is not ideal and involves overestimation or underestimation of biofilm biomass amount caused by a washing step. Therefore, in order to minimize the measurement error, our experiments were repeated three times (so-called technical repeat); in each of them, the optical density was read from 6 wells of the microtiter plate, which gave a total of 18 so-called biological repeats for each tested sample.

In the manuscript, Figure 7 provides sample photos of the investigated biofilm cultures. However, our goal was to visualize morphological changes in bacterial cells under antibacterial agents. We agree with the suggestion that it would be good to carry out continuous microscopic observation of biofilm development and make photographic documentation that would enrich the presented results.

REVIEWER COMMENT:  The coloring of the diagrams is a bit extravagant for publications.

ANSWER: The coloring of the diagrams has been changed.